# Adapting Christianity to Hakka Culture: The Basel Mission's Activities among Indigenous People in China (1846–1931)

**Lei Li**

College of Foreign Languages, Nankai University, Tianjin 300071, China; leelei@nankai.edu.cn

**Abstract:** The Hakka are a branch of the Chinese Han people, who immigrated from central China to Kwangtung (Guangdong 广东) Province. They have their own cultural norms in terms of language, lineage, distribution of work roles and status of women. The trans-national Basel Mission was headquartered in the Swiss city of Basel, near the Swiss–German border. The Basel Mission was distinguished among the missions to China by its rural Hakka Christian community. This article sets out to illustrate how the Basel Mission supported and maintained the rural Hakka Christianity community by integrating Christianity with Hakka cultural precepts. Previous Christian historiographical research has generally chosen not to emphasize Hakka cultural beliefs and practices. Examining the activities of the Basel Mission from the perspective of the indigenous Hakka culture, this article aims to enhance our understanding of the cultural precepts of receptors to shape the global enterprise of missionary society.

**Keywords:** the Basel Mission; Hakka Christianity; Hakka cultural precepts; Christianity in China





## 1. Introduction

China was already recognized as one of the world's two main mission fields by the time of the 1910 missionary conference in Edinburgh (World Missionary Conferrence 1910). In *the Reference Guide to Christian Missionary Societies in China*, Tiedemann (2009), lists 254[1] Protestant missions in China, and mentions a total of approximately 45,000 to 50,000 Protestant missionaries in China between 1807 and 1952. The Basel Mission (BM) sent more missionaries to China than other German [speaking] missions (Sun 2002). It was a trans-national Swiss-German Protestant missionary society, with its headquarters in Basel, a Swiss city on the border with Germany. However, the majority of its missionaries, as well as a large proportion of its financing, came from Germany, especially from the southwestern German state of Baden-Württemberg. The history of the BM in China goes back to 1847, as it came to an end in 1951. Its mission fields were in the highland regions of the Kwangtung (Guangdong 广东) Province and the lowland areas around Lilang (today's Shenzhen 深圳). The BM worked mainly among the Hakka people, who are known in the West for their revolutionary figures, an example being Hong Xiuquan. Indeed, the best-selling 2019 German historical novel, *God of the Barbarians*, begins with the journey of a BM missionary to join Hong Xiuquan's Taiping rebellion (Thome 2019).

The BM's language of correspondence was German. Most of the BM archives pre-1914 are in "Kurrentschrift", a type of German handwriting based on late mediaeval cursive writing. This has proved a hurdle for many researchers trying to conduct research into the history of the BM in China. Chinese scholars, such as Chi Kong Li, Wing Sze Tong, Huirao Cai, Jianbo Leng, and Yinan Luo have studied the history of the BM in China.[2] However, their research either examines the evangelical activities of the BM chronologically or focuses on the independence of the BM's Chinese Church. No existing research explores the relationship between Hakka culture and the BM's activities.

Western scholars have attached considerable importance to the meeting of and compromises made between the Hakka people and the BM. Jessie Gregory Lutz, an American

scholar of the history of Chinese Christianity, has pioneered research into the BM in China, describing how their understanding of Hakka family lineage underpinned the BM's early conversion of Hakkas to Christianity (Lutz and Lutz 1998, pp. 35, 38, 72). Her writings give insight into the geographical, social and cultural characteristics of the BM-Hakka mission fields, drawing on the autobiographies of eight Hakka Christians. Thoralf Klein focuses on how Hakka Christians reacted to the BM from the time of the late-Qing dynasty to the rise of the Chinese communist party (Klein 2002, pp. 31–40). However, neither Lutz's nor Klein's writings address how the Hakka culture shaped the activities of the BM in China. This unexplored question is of significance for two main reasons. Firstly, the Hakka Christian community was regarded as the "focal point of a new identity" (Lutz and Lutz 1998, p. 261) and "they [the Hakka Christians] reconciled Christian and Chinese facets of identity" (Constable 2013). To understand the origin of the "focal point" (as mentioned by Lutz), how the BM adapted its activities to Hakka culture needs to be fully understood. Secondly, our knowledge of indigenous practices enhances our ability to understand how the institutional culture of the BM was, in turn, influenced by Hakka culture.

This article sets out explore how the BM started its activities in China and adapted its activities to Hakka culture. The research draws on the BM archives[3], Chinese publications, and the data collected by the author during her field work to Hakka regions in Kwangtung (Guangdong) Province in 2018. It will describe how the BM began its mission venture in China, and move on to describe the cultural characteristics of Hakka people and how they influenced the BM's missionary activities and approach.

## 2. The Basel Mission in China and Its Evangelization of the Hakka People

Karl Friedrich Gützlaff (1803–1851) was a key figure in the BM's initiating of its mission enterprise in China. He was the strongest promoter of German missionary activities in China and one of the most influential figures in the history of Sino-German relations in the 19th century. However, his relationships with the BM were more complicated than with other German missionary societies.

Gützlaff's findings about China excited the BM's interest in working in China. In 1843, Gützlaff established the Chinese Union (Chinesischer Verein), whose Chinese name "fuhuanhui 福汉会" meant that Chinese Han people would be blessed after their conversion to Christianity (Y. Wang 1997). He recruited eligible Chinese to preach Christianity in the Chinese interior after their own conversion. The Chinese Union had over 100 members (Steiner 1915, p. 63). Gützlaff had already asked the BM to send missionaries to China in 1839 (Steiner 1915, p. 62). Since China was still closed to Christian missionaries at that time, the BM did not respond to Gützlaff's request to send missionaries. The claim that "China was open to evangelism", however, was widely heard in western missionary societies after the 1842 Treaty of Nanking. The BM thought it could not "get away from the requirements of the missionary friends at home [who wanted to respond to] the enthusiastic invitation of Gützlaff" (Steiner 1915, p. 62). It therefore sent Theodor Hamberg and Rudolf Lechler as missionaries to China in 1846. Hamberg was to be responsible for 3–4 million Hakkas, and Lechler was to be in charge of the Hoklo people in the coastal region, which extended as far as Fukien (Schlatter 1916, p. 276). In order to launch this plan, the BM Committee decided to grant 5000 Swiss francs for two years' work in China (Komittee Protokoll 15. Juli 1847, p. 164).

After the arrival in Hong Kong of Hamberg and Lechler, Gützlaff sent Hamberg to work among the Hakkas in Lilang, and Lechler to evangelize to the Hoklos in Yanzao (yanzao 盐灶) of Shantow (shantou 汕头), respectively. Each man had an assistant, who was a member of the Chinese Union. However, Lechler and Hamberg realized that China was not as open to evangelism as Gützlaff had reported in Europe, and that the Chinese Union's achievement was more fantasy than fact.[4] It seemed that the employees of the Chinese Union were actually deceiving Gützlaff. Their reports of preaching activities and successful conversions were apparently often only written with a view to earn money.

In 1849, Gützlaff went to England and mainland Europe to lobby for more support. He distributed responsibility for different parts of China to mission allies in Britain, Holland, and Germany. It was Theodor Hamberg and the BM Committee who exposed the deception of the Chinese Union while Gützlaff was gaining widespread support in Europe. Hamberg led an investigation into the Chinese Union in Hong Kong. Members of the Chinese Union confessed that they had engaged in deception. They had either invented the names of converts or sold the brochures they had been given free of charge by Gützlaff in order to make money. Some of them even spent the income on opium, it appeared. Hamberg wrote to the president of the BM, Wilhelm Hoffmann, in March 1850, saying that it looked as if Dr. Gützlaff had made an agreement with the Chinese for the purpose:

> I will give you money and opium in exchange for your telling certain lies. This way, I will make a name for myself, and you will have plenty of money. He [Gützlaff] is only too aware of the kind of mission he is running, and we [the BM] should not be following his example.[5]

After Hamberg's letters arrived in Basel, the director of the BM, was so disheartened, that he immediately decided to prevent any further publications of Gützlaff's reports (Schlatter 1916, p. 278). The BM Committee tried to prevent any further financial support going to Gützlaff. Because of Hamberg's claim and the BM's leadership questioned Gützlaff and the Chinese Union, support for Gützlaff's Chinese Union in German-speaking Europe came to an immediate end.

As a result of the uncovering of the deception of the Chinese Union, missionaries of the BM chose to work independently. In 1852, the BM mission committee sent a third missionary, Philipp Winnes, to China. At the end of 1852, Lechler, Hamberg and Winnes decided to establish a mission house in Hong Kong and expand from there to the Chinese mainland for the evangelization of the Hakka people in the Chinese interior. The reasons for their decisions were as follows:

(1) A firm mission post in Hong Kong was considered an ideal information-collection point and a place where people could stay if they were expelled from the Chinese mainland. Chinese hostility towards Christianity was still widespread.

(2) BM missionaries would always feel disadvantaged, constricted and neglected alongside the [more prosperous] British missionary societies in Hong Kong. On the other hand, they were free to put down roots according to their circumstances if they worked in the Chinese mainland (ibid., p. 292). Family ties in the Chinese mainland were more favorable than in Hong Kong for the spreading of Christianity.

(3) Hamberg and Lechler's association with the Chinese Union ensured that they had good Chinese assistants, and it also helped them master the Hakka language, enabling them to work efficiently among the Hakkas.

With its focus on the Hakkas, the BM was able to focus its manpower and financial resources on efforts such as the translation of the Bible and other liturgical texts (Klein 2002, pp. 16–17, 2021, p. 173). Its evangelical work focused on developing rural Hakka Christian communities, according to *Christian Occupation of China* (Stauffer 1922, p. 353). However, the BM was not the first organization to engage in the cultural exchange with Chinese. As early as the 17th century, as Nicolas Standaert shows, the Chinese were already influencing the Jesuits missionaries, making it possible for them "to become who they became". Their significant role in Sino-Western cultural exchange was markedly influenced by the Chinese. Broadly speaking, the "corporate culture" of the Jesuits as an institution was determined by the Chinese they encountered (Standaert 2014). In a similar manner, the "corporate culture" of the BM was influenced by the Hakka cultural characteristics.

## 3. Cultural Characteristics of the Hakka People as an Ethnic Group

The Hakka people are a branch of the Han Chinese, but they were marginalised by the high Han Chinese culture. They differ slightly from the Han Chinese in terms of features, customs and religious beliefs. Luo Xianglin, recognized as the founding father of Hakkology, shows in his *Introduction to Hakka Studies* that Hakka people do, indeed, belong

to the Han ethnic group. Christiansen (1998), however, disputes not only Luo Xianglin's writings, but also his views. One of his criticisms is that Luo's research evidence is poor and unconvincing. Another is that he takes a Darwinian approach to the social and moral qualities of the Hakka, which, in his view, undermines accepted scientific standards for research because it merges ideas about how racial descent and the physical environment affect community. By contrast, in his *Constructing Subjects of Knowledge Beyond the Nation: Transcultural Layers in the Formation of Hakkaology*, Thoralf Klein echoes, and implicitly, supports, Luo Xianglin's appraisal of the Hakka's special characteristics, for example, their industriousness and frugality, their ambition and readiness for action, their self-sufficiency and the (comparatively) higher status of women (Klein 2021, p. 168).

These slight differences or cultural traits distinguishing the Hakkas from other Han people predisposed them to being receptive to the BM's evangelism. However, neither Christiansen's criticism nor Klein's agreement with Luo's interpretation of the qualities of the Hakkas is comprehensive. To achieve a comprehensive representation and interpret ethnic Hakka characteristics, we should look at both points of view. In fact, the physical environment in which the Hakka lived together with their history shaped their language, their attitude towards family lineage, distribution of work roles, and the [comparatively] higher status of Hakka women.

The Hakka language is the most important factor distinguishing the Hakkas from other Han Chinese groups[6]. Indeed, Wang Dong goes as far as to define the Hakka people as a "dialect group" (D. Wang 2007, p. 35). The number of sounds used in Hakka is 619, in Mandarin, it is 532. The number of tones used in Hakka is 6, in Mandarin it is 5 [including the "neutral" tone] (Eitel 1867b, p. 66). To some extent, their mountainous living environment isolated the Hakka and prevented them from interacting with other communities, and, indeed, the Hakka language has retained more of the characteristics of ancient Chinese than other Chinese dialects. This means that translating the scriptures and liturgical literature into Hakka was the primary means of adapting Christianity to the Hakka culture.

Hakka (kejia 客家) means "guests/migrants", reflecting how the "migrations" of Hakka cultural history shaped the very character of this ethnic group. Because of turmoil and wars, Hakka people engaged in several migrations from North China to South China over a long period of time. Most scholars of the Hakkas and Hakka history agree that the Hakka people migrated five times altogether.[7] Hakka identity and characteristics emerged over the course of these migrations (Wen 2011, p. 38). Ernst Johann Eitel, a BM missionary who pioneered Hakka studies, went as far as to compare the Hakka people with the Israelites of the Old Testament (Piton 1873–1874). In the course of their migrations, the notion of family lineage emerged as crucial to the feeling of cohesion which united the Hakkas and enabled them to overcome adversity. They preserved genealogical records showing their lineage. Notes recording the migratory history of individual families are present in many of these records. After they settled in a new place, it was their lineage that bound them together in their efforts to seize arable land from the local Puntis. A family bound by lineage usually lived together in a large Hakka "roundhouses"[8], or at least in the same village. In the latter case, the village was named after the lineage's family name. Their descendants, however, chose to move to a number of nearby villages. During their frequent migrations, ancestor cult was vital to maintain their sense of lineage. Ancestors symbolized their identity, and the veneration of ancestors was embedded in Hakka social structures, serving as the lynchpin of Hakka moral values. Before embarking on a new migration, the Hakka would dig up the bones of their ancestors and carry them in clay urns to their new place of settlement. Once they had settled in a new place, they would bury their ancestors again. Respect for ancestors was also reflected in the structure of their houses. A family's room commemorating ancestors was situated in the center of any Hakka roundhouse and faced the main entrance to the house (Liu 2003).

It is common knowledge that nature nurtures culture. The mountainous landscape of Hakka country in the northeastern Kwangtung (Guangdong) Province had a profound

influence on the Hakka way of life (Wen 2011, p. 239). Essentially, where there were mountains, there were Hakkas; all Hakkas lived in the mountains. It was said that, for every eight mountains, there was only one piece of arable land. This patch of arable land was not usually sufficient to provide a living for a whole household, so the men of a household had to seek their fortunes elsewhere. The adventurous Hakka spirit which encouraged them to leave their home territory was unique (Luo 1992, p. 106). The typical pattern in Hakka households was that men would work in Hong Kong, or even in southeast Asia, while Hakka women stayed at home to tend to the farming and the family.

This pattern meant that Hakka women enjoyed a higher social status in Hakka communities than their counterparts in other Chinese ethnic groups. The social position of Hakka women distinguished them from the Puntis and Hoklos women, who lived nearby (Eitel 1867a, p. 97). Hakka women were almost as active in outdoors as Hakka men. For example, they carried heavy loads to market, took hay to the brick-kilns, and grew rice. For heavy work, they wore a "Hakka hat". They would spin yarn or make bamboo fans to sell when money was scarce. In Hakka society, women who could not farm were looked down upon (Luo 1992, p. 107). Foot binding, common elsewhere for women of status in Chinese society was unknown in Hakka culture, where women were valued for their practical skills. Because they needed to do heavy work, Hakka women did not engage in foot binding, which meant that their feet were allowed to grow to a natural size. Hakka women were expected to work in the home and in the field, and they earned respect through hard work. An elderly Hakka woman was called "po tai (婆太)" and her birthday was celebrated by a family banquet gathering. The "po tai" in question would wear special embroidered clothes and a phoenix coronet, and was afforded respect from the whole extended family at the celebration. This was quite remarkable in the nineteenth and early twentieth centuries, when Chinese women were generally still subject to what would nowadays be regarded as feudal ethics.

## 4. The Adaptation of the Basel Mission Activities to the Hakka Cultural Context

Corresponding to the Hakka cultural traits mentioned in the third section of this article, this part will retrace how the missionaries of the BM adapted their activities to the Hakka culture. Firstly, it will analyze how missionaries explored the Hakka language and how they translated the Bible into Hakka. Following, the creation of the rural Hakka Christian community based on the Hakka lineage and ancestor worship will be elucidated. Thirdly, it will explain how the BM localized their activities according to the Hakka work role of distribution.

### 4.1. Hakka Language Appropriation and the Hakka Bible as Basis to Adapt Christianity

Not knowing the Hakka language was an obstacle to the BM missionaries' work, their mother tongue being German, which belongs to the Indo-European group of languages. Mastering Hakka was key to the BM missionaries' efforts to adapt Christianity to Hakka culture. In the 1910s and the 1920s, they began learning Chinese as a first step towards learning Hakka. According to the BM magazine, *The Evangelical Messenger of the Heathens* (*Der Evangelische Heidenbote*), missionaries had to do a Chinese course before embarking on learning Hakka. The language course of 1915 is one example. BM missionaries were required to attend a two-and-a-half-month Chinese course in Pforzheim, Germany, before going to China. Gustav Adolf Gussmann, who had worked in China for 38 years (1869–1908) and his wife, were the tutors for this Chinese course. The course was held four times a year, and records show that young BM missionaries were impressed by Gussmanns' extensive knowledge of Chinese.

Missionaries began their Hakka language learning in China. The BM committee did not expect them to become Sinologists, but they had to demonstrate a good command of the spoken and written language. During their first two years in China, their main task was to learn Hakka from indigenous language teachers. Dai Wenguang, one of the first Hakka Chinese evangelists, taught Hakka to Lechler and Winnes, for example

(Eppler 1900, p. 218; Lutz and Lutz 1998, p. 58; Chappel and Lamarre 2005, p. 22). New missionaries would have annual language tests (Verordnungen für Mission, Gemeinden und Kirchen, BMA-09.1). The Hakka-German dictionary produced by Rudolf Lechler, Theodor Hamberg and their Hakka assitants was the main reference work for anyone learning Hakka. The first BM missionaries to China, in particular Lechler, compiled Hakka language textbooks and dictionaries which used a romanized rendering of Hakka words. This enabled missionaries to learn Hakka through a familiar script. In addition, BM missionaries had to learn about Hakka religious beliefs, and Hakka customs and traditions. Hakka proverbs proved a good basis for gaining insight into the Hakka mindset. BM missionary, Friedrich Lindenmeyer, who worked in Lilang and Kayintschu for 18 years (1901–1919), collected 2603 Hakka proverbs and put them into Romanized Hakka so that other BM missionaries could familiarize themselves with Hakka culture. Some proverbs, such as "a good mother gives birth to good children, and good rice seeds produce good rice", reflected the way of life, belief structure, and mindset of the Hakkas. Lindenmeyer believed that, through proverbs, missionaries could thereby learn not only the Hakka language, but also gain insight into the mindset of the Hakka (Proverbs in Hakka—romanized Hakka, written by Lindenmeyer BMA A-20,22). More importantly, as Best (2021) maintains in *Heavenly Fatherland: German Missionary Culture and Globalization in the Age of Empire*, "the respect and commitment to indigenous languages encouraged German Protestant missionaries to view non-Europeans with a sympathetic mind". Learning Hakka was a means to become familiar enough with Hakka culture to be able to preach Christianity to the Hakkas.

As stated in Section 3, the Hakka language preserves some characteristics of ancient Chinese, and it is different from Chinese Mandarin in its phonology, vocabulary and syntax. Protestant missionaries, such as Robert Morrison, William Milne, and Walter Henry Medhurst, produced several Mandarin translations of the Bible in the 1840s and 1850s. China was highly literate as a society, and it was recognized that the Christian message needed to be delivered in text form to this sophisticated civilization (Irene et al. 1999). Furthermore, the Hakka Bible was vital for the successful transmission of the BM's message of Christianity to the Hakka. Jost Oliver Zetzsche concludes from his research into Bible translations by Protestant missionaries that "the most important factor in the Chinese missionary Bible translation was missionaries' changing understanding and perception of the Chinese language" (Zetzsche 1999, p. 363). Coinciding with the Zetzsche's claim, the BM's translation of the Bible into Hakka was divided into two phases which took into account the progress the missionaries were making with the Hakka language.

In the first phase, the missionaries produced a Bible using a Latin alphabet developed in the 1850s by Karl Richard Lepsius (1810–1884), a German Egyptologist, for transcribing non-European languages. Rudolf Lechler devoted many of his 52 years (1847–1899) in China to translating the Bible into Hakka (He 1946). He translated the gospel of Matthew into colloquial Hakka using a Romanized script (Schlatter 1911, p. 203). Then, he translated the rest of the New Testament into Hakka, with the help of Philip Winnes and Charles Piton. This New Testament translation was completed in 1884 (Klein 2002, p. 183). Hakka assistants, Dai Wenguang, Kong Ayun, and Li Shin-en reviewed and revised the translation (Lutz and Lutz 1998, p. 58; Chappel and Lamarre 2005, p. 22). The BM committee was against the Chinese-character version of the Hakka Bible because they regarded it as "a specifically pagan script (eine spezifisch heidenische Schrift), and the Lepsius script was preferred as a means of freeing the Hakka from the shackles of characters" (Eppler 1900, p. 230). When the BM began its missionary work in China, the Lepsius script was adapted so as to be accessible to the Hakka people. The romanized Hakka Bible was easy for Hakka people to read if they had a good knowledge of the script (ibid., p. 324). Students at the BM mission schools were able to read it within a year, even the less able students. This would not have been possible with a script dependent on Chinese characters (ibid., p. 230). A Chinese scholar of biblical translations, Zhao (2019), believes that the romanization of Chinese for biblical translations enabled missionaries to preach to the illiterate. The efforts

of the BM missionaries to create a Lepsius-Hakka version of the Bible enabled the BM missionaries to teach illiterate Hakka people to read.

However, when the BM missionaries learnt more about the culture and needs of the Hakka, they recognized the correlation between Chinese and the Hakka language and decided to create a character-script version of the Hakka colloquial Bible. The evangelical activities of the BM worked as a learning process for them. It involved not only collaboration between the Christians and the native mission workers, but also the negotiations between the BM missionaries and the mission leaders (Klein 2002, p. 37). After interacting with the Hakka for some time, the BM missionaries, Charles Piton in particular, found that it was impossible to ignore the everyday importance of Chinese characters for the Hakka. Educated Chinese used a complex character system. Both Chinese classical literature and government documents were written in Chinese characters. Although the BM mission-school graduates had gone through the Lepsius training at the BM schools, they preferred the Wen-li version of the Bible, which was in classical Chinese and used Chinese characters (Lutz and Lutz 1998, p. 234). BM missionaries complained constantly about the committee's rejection of Chinese characters, which forced the committee to give into the Chinese [Hakka] demands in the summer of 1877 (Eppler 1900, p. 231). Lechler and his colleagues were eventually permitted to publish materials using Chinese characters. Meanwhile, translating the Bible into colloquial Hakka using Chinese characters was possible because Piton had succeeded in using them to transliterate colloquial Hakka. People only had to know 3000 (not 6000) characters to read the transliterated colloquial Hakka New Testament (ibid., p. 324). For those colloquial Hakka words for which no characters existed, he "adopted either unofficial characters known through publications in colloquial Cantonese, or used characters with an identical, or similar sounds to the words in question, then added the character mouth radical '口' to the left" [of each character to indicate that this related to the pronunciation] (Lutz and Lutz 1998, p. 234; Chappel and Lamarre 2005, pp. 324–26). Other missionaries who worked among Chinese minority groups, such as the Jingpo people, the Miao people, and the Yi people, made a significant contribution by creating characters especially for these minorities in their translations of the Bible (Zhao 2019, p. 163). The BM missionaries created many characters in their translations into Hakka and left a valuable linguistic corpus for future research into Hakka. A complete colloquial Hakka Bible using Chinese characters was first published in 1916 (Klein 2002, p. 182).

As indicated above, moving from a romanized version to a character-based version represented an attempt by the BM to adapt Christianity to the Hakka cultural context. In terms of the character script, verse 1:7 of the Acts of the Apostles (BMA II a.11) serves as a good example of how BM missionaries used the Hakka language. The style of language, the terminology, the principles underlying the translation, and the public reception would be the criteria for examining the further translation of Bible into other dialects (Zetzsche 1999, pp. 82–100). The verse in question is "it is not for you to know the times or dates the Father has set by his own authority". Most of the Hakka people were illiterate peasants, and colloquial Hakka was more accessible to them. With regard to vocabulary, the key term "father" was translated into Hakka as "a pa (亚爸)", and "you" as "ngi teu (禺兜)". To translate "teu" as "兜"is an example of how the BM went as far as "creating" colloquial Hakka words for their biblical translation. "Teu" is the plural marker in the Hakka. Basel missionaries borrowed the homophonous Mandarin Chinese character "兜", meaning "bag" or "helmet" in Mandarin Chinese to represent the Hakka "teu", although they are semantically unrelated. In Hakka Syntax, "m (唔)" usually appears as a negative particle before the verb (Zhang and Zhuang 2001). In the BM's Acts of the Apostles, "m" was added to precede "ti (知)", which means "know" in Mandarin Chinese, thus signifying "not to know". Semantically, "tsa (揸)" means "to grab something" in Hakka. Literally, "tsa khen" (揸权) means "to grip onto the authority". So, "set by his authority" was translated as "tsa khen". Thus, the whole verse was "koi teu sî khi, he a pa tsa khen loi thin tsit, ngi teu m s ti (该兜时期，系亚爸揸权来定唧，禺兜唔使知)". There were many other very Hakka expressions, for instance, HE was translated as "ki (佢)" and "why" was translated

as "tso mak kai (做乜嘅)". Nida (1964) concludes from his own Bible translation that a translation using dynamic equivalence results in natural expression and relates better to the listener/reader in terms of his or her own culture. In orientating their translations to the Hakka mindset and ensuring the naturalness of translated expression, the BM missionaries were, indeed, using the notion of dynamic equivalence to some extent. They wanted the Hakkas to relate and react to the Bible in the same way as Westerners had originally related and react to it. During the author's field work in the Hakka region[9], Wen Haiqing, a Hakka pastor in the city of Heyuan (Heyuan 河源), confirmed that the BM's Hakka Bible tended towards the Hakka dialect of Meizhou (Meizhou 梅州). Hakka Christians were grateful to the Basel missionaries because they "helped to create a unified, standardised 'beautiful Hakka church dialect', so that all Hakkas [irrespective of their dialect] could communicate with one another" (Constable 2013, p. 36).

### 4.2. Hakka Family Lineage and the Creation of a Rural Hakka Christian Community

As indicated in Section 2, the BM's main aim in China was to create a rural Hakka Christian community. Understanding the nature and importance of family lineage had proved vital for the earlier Jesuit missionaries, and also proved vital for the later Protestant missionaries in breaking down any resistance among the native population to embrace Christianity. The BM's rural Hakka Christian community was underpinned by attention to and recognition of the importance of Hakka family lineage.

Family lineage and village communities were important as social networks for creating Catholic Christian communities in the earlier period of the Qing dynasty (Zhang 2021). Hakka people typically lived in rural areas, and family lineage was an essential element in their self-identity. Around 1850, the BM missionaries experienced a spate of robberies and hostility from the local residents in the lowland regions in the Kwangtung (Guangdong) Province. As a result, these missionaries fled to Hong Kong. Jiang Jueren, a former member of the Chinese Union, suggested to Lechler that his hometown of Lilang would be a suitable place for BM evangelical work and, indeed, a mission station (Zhong 1894). He "repeatedly expressed the wish that a house could be rented for me [Lechler] in their village of Lilang" (Luo 1967, p. 7). Jiang played an important part when the BM began its work in the lowland regions. The Lin family, one of the Hakka Christian families which influenced Chinese Christianity and modern Chinese education, came from Pukak (buji 布吉), a subsidiary station to Lilang (ibid., pp. 5–6). Lin Zhengao was baptised by Hamberg, and the members of his family were the earliest Hakka Christian converts in Pukak. His son, Lin Qilian, graduated from Lilang Seminary, and subsequently devoted his whole life to evangelical work with the BM. Lin Shanyuan, a member of the third generation of the family, became the pastor of the local Hakka church.

Similarly, the family lineage of another Hakka catechist, Zhang Fuxing, helped the BM to overcome local hostility to establish itself in the highland region of northeastern Kwangtung (Guangdong) Province. In 1852, Zhang Fuxing returned to his hometown of Tschongtshun (zhangcun 樟村), to work as an evangelical preacher in 1852. Zhang Fuxing's extended family lived in nearby villages. Tschongtshun became the first BM mission station and base in the inland Hakka region of Kwangtung (Guangdong) Province. Zhang Fuxing encouraged a family member, Xu Fuguang, to convert to Christianity. In turn, Xu helped Zhang to gain access as a preacher to his (Xu's) village, and Xu sold Christian literature to his brother-in-law, Lai Xinglian. Subsequently, Zhang Fuxing, Xu Fuguang and Lai Xinglian became leading itinerant preachers for the BM in inland Hakka areas. Today, the descendants of early Hakka Christian families from the region still lead local churches. For instance, the sixth-generation descendants of Zhang Fuxing lead local churches in Tschongtshun.[10]

Tschongtshun became the centre for the BM's evangelical work in northeastern Kwangtung (Guangdong) Province. From 1926 on, Laolong (laolong 老隆), on the East River, served as the BM's local headquarters (Witschi 1965). Thanks to the family connections of the Hakka catechists, the evangelical work of the BM in the inland Hakka areas spread

to both the lowland (Unterland) and the highland (Oberland) regions. The lowland regions were made up of Lilang, Pukak, Longheu (langkou 浪口), and Tschonghangkang (zhangkengjing 樟坑径), which were near to Hong Kong and found along the coast of Kwangtung (Guangdong) Province. The highland regions were the inland mountain Hakka regions of northeastern Kwangtung (Guangdong) Province.

The connections produced by Hakka family lineage, so characteristics of the Hakka way of life, were responsible for the spread of Christianity. However, Hakka family ancestor worship stood in direct contradiction to Christian monotheistic beliefs and doctrines. In Corinthians 10:20, Christians are urged to reject idolatry. Because "pagan sacrifices are offered to demons, not to God, and I do not want you to be participants with demons. You cannot drink the cup of Lord and the cup of demons too; you cannot have a part at both the Lord's table and the table of demons" (Hong Kong Bible Society 2012). Apart from the biblical decree, George Minamiki attributes the predicament of missionaries to something else. Missionaries came from a world where there was an enormous, and at times an unbridgeable, separation between the living and the dead and between the sacred and the profane (Minamiki 1985, p. 11). In the Hakka cultural context, how to justify the ancestor cult as bridge between descendants and forefathers was a fundamental question for the BM missionaries.

The scriptures decree that Christians must not worship "demons" or partake of food deriving from sacrifice. The fundamental problem was the symbolic significance of objects and gestures involved in rituals (ibid., p. 206). In the 17th century, Matteo Ricci, a famous Jesuits missionary to China, maintained that, in the cultural context of China, rituals associated with ancestor worship were indicative of reverence for those ancestors and the sacrificial food was an expression of this reverence (Ricci and Trigault 2019). For the Chinese, the issue was whether those who converted to Christianity were also expected to adopt Western culture (Mungello 1994, p. 3).

Faced with this apparent incompatibility, the BM decreed that Hakka Christians should not put bread, silver coins or other objects into coffins, and that a funeral should not be used as an excuse to organise a larger meal (Verordnungen für Mission, Gemeinden und Kirchen, BMA-09.1). However, as discussed in connection with the translation of the Bible, the BM's evangelisation of the Hakkas was a learning process. Later, the missionaries, Heinrich Ziegler and Johannes Dilger noted that ancestor worship was significantly more complex when it came to the Hakka cultural context. They made great efforts to interpret the symbolic meaning of the objects and gestures as used on specific occasions in ancestor worship rituals, and the BM committee analyzed their findings in detail in order to draw up a code of behaviour for Hakka Christians.

When there were funerals or banquets to celebrate the birth of boys, Hakka Christians were permitted by the BM to worship their ancestors (Komittee Protokoll 29. September 1899). The BM committee made a clear distinction between ancestor worship and idolatry. Filial piety as reflected in ancestor worship was, according to the BM, a social virtue that gave stability and permanence to Hakka families and recognized Hakka family lineage. The BM committee thought that, if they banned ancestor worship entirely, the Hakka would be reluctant to convert to Christianity. The BM committee therefore came up with a compromise interpretation of ancestor worship vis à vis biblical doctrines, which was the BM's way of supporting Hakka belief in family lineage. The key question for the BM committee was whether the Hakka Christians could actually rid themselves of the notion of ancestor worship. If they could, eating sacrificial meat during festivities connected with funerals meant that they were simply enjoying a meat-based dishes to which they were entitled by virtue of lineage affiliation. Those Christians who had not been present at the graveyard (i.e., had not engaged in ancestor worship) during a funeral were permitted to eat the sacrificial meat from ancestral property and also the meat from animals slaughtered in honour of the ancestors. Attending a family ritual and traditional festive gatherings enabled Hakka Christians to maintain their family affiliation and share in their banquets connected with family lineage. All this means that the BM were adapting the doctrines of

Christianity to Hakka culture. In other words, the BM had interpreted Christian thought in such a way that Hakka Christians could avoid any identity conflict by the BM's adaptation of Christianity to their culture. Hakka Christians felt they were able to maintain their Hakka identity and still practice Christianity. The BM's accommodation of the Hakka notion of lineage had made this possible.

*4.3. Hakka Work Role Distribution and the Sustainability of the Rural Hakka Christian Community*

Covell (1995) concludes from his case studies of the Christians among the Chinese minority that "a missionary must take people to where they are in their understanding and their life situations and help them move along to a point where the gospel message and its lifestyle make sense". Teaching the Hakkas how to reconcile their lives with the message of the Bible came second only to creating a rural Hakka Christian community. In other words, the sustainability of the rural Hakka Christian community was dependent on how the BM missionaries motivated the local residents to interpret their lives according to the precepts of Christianity. As analyzed in Section 2 above, Hakka work role distribution prescribed that the husband should try to find work and earn for the family in Hong Kong or overseas, while his wife should stay at home and took care of their extended family and the farming of their land. The following aims to set out how the BM made use of this work role distribution to sustain the Hakka rural Christian community.

Elisabeth Oehler-Heimerdinger's "Bible women" (see below) in Tschonghangkang is a possible illustration of how traditional Hakka work role distribution influenced the BM's evangelical agenda. Elisabeth Oehler-Heimerdinger was the "mission bride"[11] of Wilhelm Oehler, the inspector of the BM to China after working in mission station Tschonghangkang. From 1909 to 1920, the couple worked together in Tschonghangkang. By then, every extended family household in Tschonghangkang had one or two men working in port cities in America, Australia, Britain, and Indonesia. As a result, many wives were acutely lonely, and some were even abandoned by their husbands. In such writings of hers as *What Awaits the Ho Moi: The Fate of Chinese Women (Der Weg der Ho Moi: Lebensschicksal chinesischer Frauen)*[12], she depicted the sorrowful lives of Hakka women. In addition to taking care of the missionary household, missionising to women and supervising the activities of the Bible women were the most important parts of Oehler-Heimerdinger's work (Oehler 1959).

Oehler-Heimerdinger organised local women, creating an "association of young women and girls". These women and girls were known as the "Bible women". With the help of her Hakka assistant, she trained the "Bible women" in Christian storytelling and preaching. On behalf of the BM, the "Bible women" would contact the home-bound Hakka wives. Carrying everything they needed in large, brightly coloured cloths, the "Bible women" went to non-Christian villages, trying to talk to the women who lived there. When Hakka women complained about their concerns, for example, their husbands' infidelities, that their sons lived abroad, or expressed their concern because they had no sons, the "Bible women" were adept at talking to them and providing them with succor and advice based on their own experience (Oehler-Heimerdinger 1925). The "Bible women" would give them comfort, recounting the story of Adam and Eve, because it seemed that Hakka women were quite interested in it. From 1883 to 1902, the number of converted Chinese in Tschonghangkang amounted to just 24. In 1913, a total of 129 adults and 37 children were baptised in Tschonghangkang (Schlatter 1916, p. 386). Oehler-Heimerdinger's evangelical strategy was universal among the BM female missionaries in the mountainous Hakka regions. Gertrud Schaeppi, a BM female missionary was responsible for the itinerant preaching to the Hakka women scattered in the different villages in Lenphin (lianping 连平). "Bible women" there were still ill-educated. Further, they themselves could not read, and no one taught them to do so (Schaeppi 1943, p. 70). Schaeppi organised a group of "Bible women" modeled on Oehler-Heimerdinger's to teach Hakka women in Lenphin to "pass on what they have received" (ibid.).

In the general Synod of the BM in China in 1924, BM missionaries decided to formally set up a Bible Women's School in Tschhonglok (changle 长乐), which was built in 1925

(Ärtzlichen und Frauenmission BMA-3.28). Hakka women aged between 35 and 45 could register at the school. The Hakka focus on the domestic environment inspired the BM to designate a "house mother" (Sill 2010) as the central symbol of the BM's Christian womanhood in the Hakka context. Relying on the Bible and church history, Chinese assistants, Wan Enhong, Chen Tianle and Zhang De'en, edited the pamphlet called *Nvtu Jing*《女徒镜》(*Mirror for the Female Disciples*) in colloquial Hakka with a view to cultivate the notion of the virtuous "house mother". It was a guideline according to which Hakka women could conduct how they should behave on a daily basis (BMA. III. c. 51, p. 2).

These guidelines made clear that the BM's work among Hakka women was designed to develop the profile of domestic role of women. The BM's work among Hakka women was different from that of US missions in China. US female missionaries in China in the 20th century encouraged Chinese women to rebel against doing housework (Hunter 1984). However, through the *Mirror for the Female Disciples*, the BM encouraged Hakka women to be good wives. It included twelve doctrines. Four of the doctrines focused on how to be a good Christian, and the others pertained to family life. In the doctrines relating to family life, Hakka women were encouraged to respect and love their husbands, treat their parents-in-law with due respect, raise children in a loving environment, try their utmost to help anybody visiting from afar (BMA. III. c. 51, pp. 3–8). Hakka women were being encouraged through the BM's "Bible women" and the *Mirror for Female Disciples* to play a more significant role in both the household and the community. In this way, the BM's recognition and appreciation of the Hakka women's situation, as well as the conversion of Hakka women to Christianity, helped to sustain the notion of the Hakka rural Christian community.

In terms of the male role, Hakka men were accustomed to seeking work away from home (e.g., in Hong Kong, the USA). From the 1850s, there was a cooli trade between China and Cuba. This resulted in widespread migration among people from the coastal areas of southeast China to overseas destinations in search of well-paid work. The BM helped Hakka Christians to emigrate to the West Indies, Guyana, California and Australia, especially to the North Borneo and the Sandwich Islands which now belong to the Hawaiian Islands. BM missionaries, Rudolf Lechler in particular, maintained close contact with Hakka Christians scattered all over the world. BM Hakka rural Christian communities transformed into "diaspora communities" to relocate overseas.

In 1859, 67 Hakka Christians from Hong Kong and 10 from Lilang emigrated to British West Indies. The sugar plantations there were short of workers following the abolition of slavery, and British immigration policy favoured employing Christians and their families (Schlatter 1916, p. 308). As a result of this widespread emigration from China, the BM lost a considerable number of Hakka converts to Christianity, having renounced ancestor worship and what the BM saw as idolatry. North Borneo came to be the most important among the "diaspora mission fields" of the BM. Around 1880 and 1890, BM Hakka Christians arrived in considerable numbers in North Borneo as free labour. As the British Society for Propagation of Gospel had appointed a reverend missionary to take care of the English-speaking people in North Borneo, the BM initially put the care of the Hakka emigrants to him in his hands. Because of the increase in the number of Hakka emigrants and their spiritual needs, the BM developed its own Christian communities in North Borneo. In 1883, the BM was "the first" mission to open its own church for the Chinese Christians in Kudat [in North Borneo] (Wong 2000). Around 1903, three quarters of Chinese Christians in North Borneo were associated with the BM (*Der Evangelische Heidenbote* 1903, p. 89). In 1906, the BM sent missionary Wilhelm Ebert to establish a mission station in Sabah, North Borneo (Schlatter 1916, p. 420). By about 1908, some 800 Christians were settled in six communities, and new churches were built at a cost of 4000 US dollars (ibid.). Most of the Chinese living in Pontianak and Singtawang in West Borneo were Hakka people. The BM took over responsibility for the care of the Chinese communities in Pontianak and Singtawang after the sudden departure of the former (Methodist) missionaries in 1933 (*Der Evangelische Heidenbote* 1939, p. 18). Hans Bart, a BM missionary from Bern, Switzerland, worked

in these communities with his wife, supported by two Chinese pastors, Lo Schau-on in Pontianak, and Lo En-zhu in Singkavang (ibid.).

The Sandwich Islands of modern-day Hawaii were also attractive to the Hakka Christians. Hakka emigrants commonly worked as plantation workers, cooks, laundresses and nursemaids. The Hakka emigrants to the Sandwich Islands were unlike their compatriots in North Borneo in that they took the initiative to establish their own churches. Some of the emigrants were Lilang Seminary graduates and staff from the BM mission stations in China. A former graduate who worked as preacher in Kohala in the northwestern part of the Sandwich Islands wrote to Lechler in 1879, "I have been in Kohala for eight months, and there are 26 Chinese Christians here . . . my work has not been in vain for there are seven candidates for baptism" (*Der Evangelische Heidenbote* 1879, p. 52). Chinese Christians in Honolulu raised 5250 US dollars and established a church in 1881 (Lechler 1887, p. 21). An elder and the deacon of this church were BM Hakka Christians. The elder was Li a Tschhong, and the son of a BM Hakka deacon in China called Li Tschin Kau. He had had a British education in Hong Kong and was appointed by the Hawaiian government as a Chinese court interpreter in 1883 (Lechler 1887, p. 22). Lam en Luk was the deacon in question, and he had worked at the Chinese mission station of Tschhonglok.

Emigrant Hakka Christians maintained their contact with the BM in China as follows. A large number demonstrated their loyalty to the BM by sending money to Hakka churches in China on an annual basis. Hakka Christians in Honolulu collected 150 US dollars and asked Tschinhin-Si, a Hakka Christian, to forward the money, together with a letter, to Lechler when he returned to China in 1879 (*Der Evangelische Heidenbote* 1879, p. 52). In the year of Lechler's departure from the Sandwich Islands in 1886, the money they sent to the BM mission fields in China even amounted to as much as 2000 US dollars (Schlatter 1911, p. 185). Ku(Goo) kim, an elder of the Chinese church in Honolulu, returned to his Hakka home of Meizhou in 1892 to build a school and a church (Lutz 2009, p. 145). BM Christians in North Borneo tried to help needy people in China, even though their business interests had suffered significantly from the decline in rubber price and the trade standstill following the Sino-Japanese War. In 1939, a theatre performance was organised at Singkawang's playhouse in North Borneo, and almost 10,000 Swiss Francs were raised in a single evening (*Der Evangelische Heidenbote* 1939, p. 18). In fact, BM's emigrants were still dependent on the BM missionaries for spiritual advice, and they wrote to Lechler to share the joys and regrets about their emigrant life. One emigrant to Georgetown in California went as far as writing to Lechler to update his life and ask whether he should move to the Sandwich Islands in order to preach there (*Der Evangelische Heidenbote* 1881, p. 86). Hakka Christians from Hawaii invited and paid for the Lechlers (husband and wife) to visit them during their 1887 furlough, and 80 Hakka Christians gathered to welcome them (Schlatter 1911, p. 184). The BM Seminary established at the mission station in Lilang sent graduates to overseas Hakka churches. Marie Lechler, Rudolf Lechler's wife, had founded a girls' school in Hong Kong, and some of its graduates married Hakka Christians. For example, one graduate of Marie Lechler's girls' school married Lam en Luk, the deacon of the Chinese church in Honolulu. The Lechlers even brought two graduates, Fa Yung and Men Yin, who were engaged to Hakka emigrants in Hawaii, on their furlough visit in 1886 (Lechler 1887, p. 2; Lutz and Lutz 1998, p. 145). One of Marie Lechler's students who had emigrated to Honolulu recalled in 1931 how Lecher and his wife saved her from a Hong Kong brothel at the age of 10 and took care of her (A Letter from Honolulu, 20 Oct., 1931. BMA-10.38.13). It is clear that there was a good relationship of trust between BM Hakka emigrants and Lechler.

Considering the above, the author demonstrates in this article that BM missionaries were flexible enough to adapt to the Hakka mindset and Hakka beliefs, and reached out to Hakka emigrants also. In his research into the Hoklos, Christian emigrants to Southeast Asia, Zhu Feng finds that the Methodist Episcopal Church provided support for Hoklo Christians to enable them to emigrate to Southeast Asia and find somewhere else to live when their home was under threat (Zhu 2009). Similarly, the BM supported Hakka people to emigrate from their homeland to overseas and to 'export' their Hakka lifestyle and beliefs.

Most importantly, Christianity helped Hakka emigrants to integrate with local Christian societies in their destination countries (Constable 2013, p. 201). Even if emigration marked out this ethnic group, Hakka Christian communities overseas offered a sanctuary for new emigrants from the Hakka homeland, who had to become accustomed to completely new surroundings.

## 5. Conclusions

Combining findings from German and Chinese materials with data from the author's field work, this article follows the emergence of the BM mission enterprise in China, and it gives a holistic view of how the BM adapted its activities and attitude to take account of Hakka culture. As Jessie Gregory Lutz maintains: "on the fringes of the Confucian mainstream, the Hakka were often considered more open than other Han Chinese to heterodox teachings: to popular and Buddhist sects, to Christianity, and, indeed, Taiping Christianity" (Lutz 2009, p. 142). Examining the BM's activities among the Hakka, the author agrees with Lutz in part. Both parties (the BM and the Hakkas) were constantly at pains to accommodate each other. It was a matter of constant negotiation for both parties, and very importantly, BM missionaries learnt about Hakka culture and adapted Christianity to it. They responded to the culture of the Hakkas and created a unique rural Hakka Christian community. Their flexibility in adapting their evangelical activities can be explained by the characteristics of the BM itself. The BM was a mission which targeted people from the lower social classes (Tong 2002). Before the World War I, 60 to 70 percent of the BM missionaries came from Baden-Württemberg, in southwestern Germany (Jenkins 1980). Most of these German missionaries came from a traditional village background (Jenkins 1989); they were farmers and craftsmen before they embarked on missionary work (Schlatter 1965, p. 18). This meant that there was considerable common ground between typical BM missionaries and the Hakkas they preached to. This gave them a natural empathy with the mentality and way of life of the Hakkas. Through Hakka emigration, Hakka Christianity then expanded to places such as North Borneo, the Sandwich Islands, California, and Australia. The relationship was truly interactive, each party benefiting from and accommodating the other, and the culture of the BM and its global mission enterprise was enriched by contact with the Hakkas.

**Funding:** This research was supported by the Fundamental Research Funds for the Central Universities (63222074) and China Postdoctoral Science Foundation (BSMS7100013).

**Institutional Review Board Statement:** Not applicable.

**Informed Consent Statement:** Not applicable.

**Data Availability Statement:** Not applicable.

**Acknowledgments:** The author owes an enormous debt of gratitude to Andrea Ryhn and Paul Jenkins of the Basel Mission Archives for all their great supports. The author also wishes to thank her supervisor, Andreas Heuser, during her two-year research at the Theological Faculty of the University of Basel. Thanks are also due to Iris Leung Chui Wa of the Tsung Tsin Mission of Hong Kong, and to the author's friends Tuo Chen and Le Zhang.

**Conflicts of Interest:** The author declares no conflict of interest.

## Notes

[1] Tiedemann does not include any Catholic missionary societies in his account. Among the 254 missions, four missions were Japanese.

[2] For more detailed research findings of Chinese scholars, please refer to the following works: Li, Chi Kong. 1993. *Symposium on Christianity and Modern Chinese Culture*. Taipei: Cosmic Light Press; Tong, Wing Sze. 2002. *Study of a Hakka Mission in Southern China: from the Basel Mission to the Tsung Tsin Mission of Hong Kong*. Hong Kong: Chinese Study Center on Chinese Religion & Culture; Cai, Huiyao. 2006. Essay on the History of Spreading Christianity in Shenzhen in the Late Qing Dynasty. *Bulletin of Historical Research*. no. 36:131–152; The Study of the Basel Mission and Guangdong Hakka Immigration to Southeast Asia. *Journal of Shantou University*, no. 06:31–34; Leng, Jianbo. 2014; Luo, Yinan. 2017. *Belief Identity and National Identity: the Independence Process*

of the Basel Mission in the Hakka Area of Kwangtung (Guangdong) Province. Ph.D. dissertation, Beijing Foreign Studies University, Beijing, China.

3   The Basel Mission Archives is located at Missionsstrasse 21, Basel, Switzerland. They contain reports, historical photographic records and maps of the BM mission fields. A catalogue of the archives is available at https://www.bmarchives.org (accessed on 10 July 2022).

4   Hamberg's battle with Gützlaff over the Chinese Union, please see (Schlyter 2008, pp. 96–138).

5   "Ich will euch Geld und Opium geben, ihr gebt mir zweckmäßige Lügen; dadurch mache ich mir einen Namen, und an Geld wird es nicht fehlen. Das sind die Grundzüge einer Mission, die er mehr oder weniger wissentlich treibt. Wir wollen nicht seinem Beispiel folgen" (Schlatter 1916, pp. 280–81).

6   The Han people are the majority in China. The concept of Han people came into being during the Han dynasty (B.C. 202-A.D.220), when China was unified. The ancestors of the Han lived in the interior of China, but they also inter-married with the neighbouring minorities on the Chinese border.

7   Rebellions and wars were responsible for the first three times. Minority groups on the Chinese border attacked central China at the end of the Eastern Jing Dynasty, and the Han people of central China fled from their hometowns and went to the region in the south of Henan and Hubei. Then, due to the Huangchao Rebellion (874–883), they migrated from Henan and Hubei to southeastern Jiangxi and southeastern Fujian. In 1276, they were forced to migrate further south several times, because the Mongols overran the whole of China. Finally, the Hakkas arrived in what is now the northeastern part of Kwangtung (Guangdong) Province and settled there, believing that the mountains would protect them from wars. This region is seen as the Hakka homeland, and the city of Kayintschu (Meizhou 梅州) is sometimes even called the hometown of the Hakkas. However, the Puntis -local to the region- saw them as intruders and the two groups were constantly at war with each other over the limited arable land available and other resources, such as eligibility to attend the imperial examination. Later, because of their increasing population and the limited arable land available in the mountain regions where they lived, a new weave of migration began. They moved from northeast Kwangtung (Guangdong) Province to Sichuan, Guangxi, Taiwan, and Hainan island. Some even emigrated overseas.

8   In Chinese, Lung Wai housing (weilongwu 围龙屋). It is circular, multi-storey, fortress-like dwellings designed for defensive purposes with walls of adobe or tamped earth which were almost a metre thick.

9   A visit to the Hakka church in Heyuan city. Date and time: 31 January 2018, 9:00–11:00.

10  The author found this out during her field work in Tschongolk on 28 January 2018. She visited the commemorative ancestral room of Zhang Fuxing and interviewed Zhang Weiwen and Zhang Renhe, who belong, respectively, to the fourth and fifth generations of Christians in Zhang's family.

11  The BM prescribed that a missionary should search for a bride and had regulations governing their marriage. A male missionary could apply to the BM committee to find a bride for him in the mission's home base. The bride could communicate with missionaries in BM's mission fields in China, India, Ghana and Cameroon by writing to them. As a pre-requisite to becoming a mission bride she had to pass the examination of the BM. Thereafter, she could travel to one of the mission fields to marry her missionary bridegroom. More information for the mission bride may be found in Konard, Dagmar. 2013. *Missionsbräute: Pietistinnen des 19. Jahrhunderts in der Basler Mission*. Münster: Waxmann Verlag GmbH.

12  For publication information for the book, see Oehler-Heimerdinger. 1932. *Der Weg der Ho Moi: Lebensschicksal chinesischer Bibelfrauen in China: Bilder aus dem chinesischen Frauenleben*. Stuttgart and Basel: Missionsverlag G. m. b. H.

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
