# Peer review of "Adapting Christianity to Hakka Culture: The Basel Mission’s Activities among Indigenous People in China (1846–1931)"

_religions, doi:10.3390/rel13100924_

Round 1

Reviewer 1 Report

Premise: My competence in German is very limited, therefore, I had been assessing the article without delving into the accurateness of German sources and their translation into English.

The article proposes an interesting outlook over the famous Basel Mission in China during the late 19th century and the early 20th century and its impact on the indigenous population there. It pays much attention to the Hakka minority and the role of missionaries in the popularization of Christianity among them. There are numerous scholars who proposed very well-established studies on the mission. Among others, the studies by Jessie Gregory Lutz (partially mentioned in the article) and Hermann Witschi are particularly valuable and helpful. Moreover, especially for the Hong Kong context, the works by Wing Sze Tong 湯泳詩 and, in part, the ones by Nicole Constable, are crucial for the understanding of the ambitions of the Basel Mission.

The author provides a large variety of interesting accounts, although sometimes quite scattered, concerning the overall mission. Given the vast literature on the topic, the reviewer would have expected a clearer analysis of the sources retrieved by the author at the Basel Mission archives. Nevertheless, some important remarks, like the emancipation of Chinese women in Hakka society, are very valuable and shall be expanded in order to provide to the reader a more profound understanding of the mission.

The mentions concerning the Hakka cultural lifestyle are very interesting, and the sub-chapter 4.1 would need to be further expanded since it deals with very important concepts related to Chinese Christianity within the Hakka communities. Moreover, the whole article shall be restructured following a much clearer analysis. I warmly invite the author to clarify, among others, the following issues:

1)    The conflict between local practices and the missionaries’ ones (p. 9). This shall be further expanded and the few mentions on Ricci and Legge are insufficient in order to provide a comprehensive analysis of this very complex phenomenon. A good starting point would be to mention at least the following references on the Chinese rites controversy: David Mungello, The Chinese Rites Controversy: Its History and Meaning, Berlin 1994; George Minamiki, The Chinese Rites Controversy: From Its Beginning to Modern Times, Chicago 1985; The Rites Controversies in the early modern world, edited by Ines G. Zupanov and Pierre Antoine Fabre, Boston 2018.

2)    The role of the Basel Mission in the production, diffusion and translation of the Bible (p. 7) in the Hakka communities. This part shall be re-arranged, and the author shall pay more attention to the various controversies related to the production of these religious works. For a more comprehensive analysis refer to Jost Oliver Zetzsche, Bible in China: The History of the Union Version or the Culmination of Protestant Missionary Bible Translation in China, London 2017.

3)    Chapter 3 and Chapter 4 shall be expanded adopting a much clearer historical and historiographical approach, including debates concerning other Protestant missions and Catholic missions, emphasizing the peculiarities of the Basel Mission. For more references (expanding it also to the Hong Kong context), see 湯泳詩, 瑞澤香江: 香港巴色會, 香港 2005 and Nicole Constable, Christian Souls and Chinese Spirits: A Hakka Community in Hong Kong, Los Angeles 1994.

4)     The Conclusion is extremely short and it shall be expanded and re-arranged.

The article does offer some important insights on the mission, but its style is extremely unclear and it will need a major English proofreading. There are several major grammatical mistakes (e.g., p. 2 “Chinese scholars [....] has researched” should be “have researched”). Moreover, sometimes Chinese terms, like in the case of p. 2 lack their pinyin transcription. Other times, like on p. 6, the Chinese provinces are written with a different romanization (e.g., “Kwangtung province” shall be either “Guangdong province” or “Kwangtung (Guangdong) province”. Given this inconsistency in the romanization, it is difficult to have a clear understanding about all the places/individuals mentioned there. This alternative romanization could be kept if a list of the terms will be provided at the end of the article, also including the pinyin romanization and the original Chinese characters. Sometimes, like on p. 7, the romanization is a bit more consistent.

To sum up, the article could be potentially an interesting contribution to the field of Christianity in China during the modern era. Unfortunately, it lacks coherence and consistency. Moreover, its style is very difficult to follow and grasp, and it will require a substantial English proofreading. The author shall not feel discouraged by this, given the important findings within the article. However, he or she will need to further re-arrange his or her thoughts including also more primary and secondary sources on the topic. As it stands now, the article cannot provide valuable insights to recent scholarship but with a major revision - including a substantial and accurate English proofreading - it could potentially be re-considered for publication.

Reviewer 2 Report

Comments of peer reviewer 1

This is an overall well-designed research paper with a clear research intention and based on solid research. What is particularly laudable is that the author combines field research with archival studies.

However, the outcome could be improved. In fact, quite exactly half of the paper is concerned with preparatory information that is available in various places (Klein, Lutz, Schlatter, a.o.) and tells the story of Gützlaff, of the first missionaries Lechler and Hamberg, of the conflicts about the Chinese Union, about the cultural characteristics of the Hakka, etc. Only the second half of the paper is really dedicated to what is more narrowly the research of the author. Within this more relevant part, insufficient thought is given to how much what is claimed as adaptation of the missionaries to the Hakka contexts is unique to the Basel missionaries and how far this reflects work modes that are commonly used by missionaries at the time. For instance, was Elisabeth Oehler-Heimerdinger’s ministry among the Hakka women something unique or was it not rather common that women missionaries reached out to rural women? What is the difference between what Oehler-Heimerdinger did to what other female missionaries did? Maybe the most interesting part, for this reviewer, was the description of the BM’s tolerance regarding ancestor worship. This could have been further explored. On the other side, the support of the emigration that was BM-internally very much debated (Lechler vs Winnes – see Lutz 2009 in this regard) could have been further discussed. Was it an attempt to protect the familial community or did it rather undermine it? While Lutz 2009 offers a good discussion of the issue, this could actually have been further deepened by archival material. How did the BM, beyond Lechler and Winnes, view the matter?

In conclusion, with all sympathy for this research and the contents, the outcome remains rather thin. Also the outcome of the field research seems rather thin. Still, I would actually encourage publication, already for the simple fact that this scholar deserves encouragement and for the interesting choice of topic and of the overall framing of the question.

Further points for consideration and further reflection:

-          The scholarly contribution of Standaert in emphasizing the impact of local culture on overseas missionaries’ modes of work could be further described. A better explanation of his contribution would also make what appears as a bit a sudden mentioning of the Jesuits better understandable.

-          The BM mission’s emphasis on language studies could be further explained with reference to romanticism and the notion of nation.

-          The use of the Lepsius system might be critically discussed as a counterargument to the author’s argument of the missionaries’ contextualization, that is, it may be seen as an example of an insufficient contextualization, yet, the author could point out that the use of the Lepsius system was in fact a point of conflict between missionaries in the field and the home committee.

Another issue relates to language and formal aspects: Assumedly written by someone for whom English is not the native language, the author shows good English writing quality. Yet, it still needs significant proofreading to avoid the plenty of linguistic mistakes. I recommend the author to seek professional proofreading. Besides, there is a large number of instances that appear sloppy – some of the points are listed below, but it is an incomplete list. These are partly mistakes, partly inexactness’s, or points that need further explanation (number referring to lines):

-          24-25: To be clearer, better write “Among the German missionary societies, the BM was the one with …”

-          33 “best-selling” rather than “best-sold”

-          74 “Chinesischer Verein”

-          98-99 – the other way round, Lechler to the Hoklo (not Haklo) and Hamberg to the Hakka

-          155: delete ‘also’

-          177: the reference to Eitel 1867a in support of the number of tones of Mandarin and Hakka is outdated. Usually, Mandarin is seen as having 4 tones. Or if you want to make use of him as a primary source, explain why you refer to him.

-          179: as conclusion of the previous sentence, Hakka should rather be an early phase, not the latest phase, of the Chinese language. Usually, Cantonese is seen as corresponding most with early Chinese.

-          222: explain ‘Lung Wai’

-          257: ‘ethnics’?

-          272ff: when was the preparatory Chinese language course in Germany introduced? From when to when was this the preparatory practice?

-          297: ‘Heavenly Fatherland’ – should, I assume, be a book title – mark it as such

-          324: why was translated into zuo ye jiu (做乜嘅) – is this correct? Does 做乜嘅 not rather mean “what”?

-          350: Should probably be Jiang Jiaoren.

-          416: quotation of Nicole Constable without reference

-          434: refer to Elisabeth Oehler-Heimerdinger by her family name, not by her given name

-          440: chinesischer Frauen

-          501: Hokol? – probably Hoklo

-          539: I don’t think that Schlatter should be listed under primary sourcesComments of peer reviewer 1

This is an overall well-designed research paper with a clear research intention and based on solid research. What is particularly laudable is that the author combines field research with archival studies.

However, the outcome could be improved. In fact, quite exactly half of the paper is concerned with preparatory information that is available in various places (Klein, Lutz, Schlatter, a.o.) and tells the story of Gützlaff, of the first missionaries Lechler and Hamberg, of the conflicts about the Chinese Union, about the cultural characteristics of the Hakka, etc. Only the second half of the paper is really dedicated to what is more narrowly the research of the author. Within this more relevant part, insufficient thought is given to how much what is claimed as adaptation of the missionaries to the Hakka contexts is unique to the Basel missionaries and how far this reflects work modes that are commonly used by missionaries at the time. For instance, was Elisabeth Oehler-Heimerdinger’s ministry among the Hakka women something unique or was it not rather common that women missionaries reached out to rural women? What is the difference between what Oehler-Heimerdinger did to what other female missionaries did? Maybe the most interesting part, for this reviewer, was the description of the BM’s tolerance regarding ancestor worship. This could have been further explored. On the other side, the support of the emigration that was BM-internally very much debated (Lechler vs Winnes – see Lutz 2009 in this regard) could have been further discussed. Was it an attempt to protect the familial community or did it rather undermine it? While Lutz 2009 offers a good discussion of the issue, this could actually have been further deepened by archival material. How did the BM, beyond Lechler and Winnes, view the matter?

In conclusion, with all sympathy for this research and the contents, the outcome remains rather thin. Also the outcome of the field research seems rather thin. Still, I would actually encourage publication, already for the simple fact that this scholar deserves encouragement and for the interesting choice of topic and of the overall framing of the question.

Further points for consideration and further reflection:

-          The scholarly contribution of Standaert in emphasizing the impact of local culture on overseas missionaries’ modes of work could be further described. A better explanation of his contribution would also make what appears as a bit a sudden mentioning of the Jesuits better understandable.

-          The BM mission’s emphasis on language studies could be further explained with reference to romanticism and the notion of nation.

-          The use of the Lepsius system might be critically discussed as a counterargument to the author’s argument of the missionaries’ contextualization, that is, it may be seen as an example of an insufficient contextualization, yet, the author could point out that the use of the Lepsius system was in fact a point of conflict between missionaries in the field and the home committee.

Another issue relates to language and formal aspects: Assumedly written by someone for whom English is not the native language, the author shows good English writing quality. Yet, it still needs significant proofreading to avoid the plenty of linguistic mistakes. I recommend the author to seek professional proofreading. Besides, there is a large number of instances that appear sloppy – some of the points are listed below, but it is an incomplete list. These are partly mistakes, partly inexactness’s, or points that need further explanation (number referring to lines):

-          24-25: To be clearer, better write “Among the German missionary societies, the BM was the one with …”

-          33 “best-selling” rather than “best-sold”

-          74 “Chinesischer Verein”

-          98-99 – the other way round, Lechler to the Hoklo (not Haklo) and Hamberg to the Hakka

-          155: delete ‘also’

-          177: the reference to Eitel 1867a in support of the number of tones of Mandarin and Hakka is outdated. Usually, Mandarin is seen as having 4 tones. Or if you want to make use of him as a primary source, explain why you refer to him.

-          179: as conclusion of the previous sentence, Hakka should rather be an early phase, not the latest phase, of the Chinese language. Usually, Cantonese is seen as corresponding most with early Chinese.

-          222: explain ‘Lung Wai’

-          257: ‘ethnics’?

-          272ff: when was the preparatory Chinese language course in Germany introduced? From when to when was this the preparatory practice?

-          297: ‘Heavenly Fatherland’ – should, I assume, be a book title – mark it as such

-          324: why was translated into zuo ye jiu (做乜嘅) – is this correct? Does 做乜嘅 not rather mean “what”?

-          350: Should probably be Jiang Jiaoren.

-          416: quotation of Nicole Constable without reference

-          434: refer to Elisabeth Oehler-Heimerdinger by her family name, not by her given name

-          440: chinesischer Frauen

-          501: Hokol? – probably Hoklo

-          539: I don’t think that Schlatter should be listed under primary sources 

Reviewer 3 Report

The article covers a very interesting topic, yet not much explored to date. The author presents the Basel Mission and the features of Hakka Christians supporting the differences occurring with respect to the general issues of evangelization among Han people. The paper is a precious contribution to the field of study on Christianity in China and gives an original perspective on a section of research rarely considered. The use of primary sources as archival material and data from fieldwork are elements that enrich the significance of the article. 

The text is well structured and gives a clear outline of the features of the BM (par. 2) and of the Hakka people (par. 3): it is suggested to give further details on the features of Hakka Christianity in paragraph 4. In particular, references or notes to Gutzlaff as a Bible translator (par. 2) and to Hong Xiuquan and the Taiping movement would be important to strengthen the discussion. Despite the frequent reference to fieldwork, it is not easy to detect what data resulted from it, except when specified in note 7: it is suggested to give further details on both the data (through the whole text) and the fieldwork (add a note in line 64: where, when, how it was carried out).

The archival material is fundamental, however little is reported on the Archive itself. A note in line 63 could describe where the BMA is, what is stored there, etc. 

The reference to Catholic missions in line 346 is confusing and not clear: give further details or delete. The same is for British Guyana only mentioned in line 519 and not in par. 4. 

In lines 330-338 is suggested to add a reference to the volume edited by Eber Bible in Modern China: the Literary and Intellectual Impact, 1999. 

Line 327: explain in a note what is dynamic equivalence.

The use of Chinese characters with pinyin or proper names of people and places (lines 183, 253 etc.) would be important to support clarity.

The addition of "Chinese Christianity" or "Christianity in China" among the keywords would widen the searchability of the article. 

Reviewer 4 Report

Review of article “Adapting Christianity in the Hakka Cultural Context”

There are some interesting aspects to this article which I would encourage the author to expand on. On the whole, I think some major revisions are acquired.

1.     There are some interesting aspects to this article which I would encourage the author to expand on. On the whole, I think she overstates the originality of her approach. Essential discussion points of this essay, such as Bible translation and work among women, are also mentioned by Lutz and Klein. As the author herself admits, Lutz writes a great deal about how lineage impacted the BM’s strategy of evangelisation. The author also fails to sufficiently take into account Klein’s argument that the BM’s strategy of evangelisation among the Hakka was the outcome of an interactive learning process (see also point 7).

2.     The discussion of Gützlaff and the beginning of BM work among the Hakka (pp. 2-4) is too long compared to the rest of the article and does not contribute much to the author’s argument. It should be left in place but cut back considerably.

3.     The author’s understanding of the Hakka as an ethnic group (pp. 4-6) is very simplistic and uncritically follows Luo Xianglin (for a methodological critique see Flemming Christiansen 1998, available as PDF at https://vbn.aau.dk/en/publications/hakka-the-politics-of-global-ethnic-identity-building, and Thoralf Klein (2021), “Constructing Subjects of Knowledge Beyond the Nation: Transcultural Layers in the Formation of Hakkaology (Kejiaxue 客家學)”, Monumenta Serica 69:1, 161-182, DOI: 10.1080/02549948.2021.1910151, as well as Chen Zhiping 1997 in Chinese). Characterising it as a “clan” or “tribe” (p. 5) is terminologically confusing, as the term “clan” is later used (p. 5 and later) to refer to Hakka descent groups. “Dialect group” would be a much better term. The claim that the Hakka language is somehow purer Chinese (p. 5) was first advanced by missionaries (not only BM) in the 19th century and is highly controversial, as is the idea that Hakka were especially revolutionary – and the evidence that Sun Yatsen was Hakka is rather weak (p. 1). Hakka women did not have their feet bound but were otherwise treated as inferior to men (however much their status may have been higher in comparison to other Han groups). The term “household mode of production” is also misleading, as it suggests that production took place within the household when the author is actually showing the contrary. The division of labour between men (overseas) and women (at home) certainly existed in specific families but it was by no means universal (many men tilled the fields), nor was it uniquely Hakka. The same division must have taken place among the Punti and Hoklo, both of which also heavily contributed to overseas migration. The author should engage with more critical literature on the matter and incorporate that into her argument.

4.     The discussion on Bible translation and the example given are very interesting and I would like to learn more about this dimension, which is one of the most innovative aspects of the article. Expanding this section will add to the originality of the research. Even so, the argument is not clear here. A translation is either “faithful” (whatever that means) or a “free translation” (p. 7); it cannot be both. The author needs to take a clear decision here. She would do well to engage with the work of Hilary Chappell & Christine Lamarre, A Grammar and Lexicon of Hakka: Historical Materials from the Basel Mission Library (2005). I notice that the phonetic rendering of the characters in this article is in Mandarin rather than Hakka – the author should use a Hakka transliteration here (using either the unpublished BM dictionaries or MacIver’s published 1905 dictionary or, indeed, Chappell and Lamarre’s work). Character especially using the mouth () radical are specially created to render dialect words in characters; this should be mentioned. I wonder if Yaba is a Chinese transliteration of the Aramaic word “Abba”, often used in the New Testament for “father”.

5.     Regarding the discussion of clans, I would prefer the term “lineage” used by most Western anthropologists with regard to China (the author uses both terms interchangeably). The claim that the BM “permitted to worship ancestors” is, as such, wrong. The BM never permitted Christians to participate in the ancestral ritual itself. The whole debate and compromise were about whether or not Christians could partake of sacrificial meat, as the author herself subsequently details. Towards the end of the 1890s, the Committee did indeed allow Christians to consume sacrificial meat.

6.     The discussion about the ways the BM mitigated the impact of migration and how it looked after Christians and congregations overseas is very interesting and I’d love to learn more about this. It would strengthen the originality of the article.

7.     In general, the author generally pays little heed to the historical evolution of the BM’s activities. The Chinese (or rather Hakka) courses in Pforzheim represent a rather late stage in the BM’s language training. By the same token, the institution of Bible women was much older than 1909, when Oehler-Heimerdinger started her work among women. The author should better contextualise the BM’s activities rather than offer piecemeal information.

8.     In terms of her method, the author emphasises the role of the BM Archives as well as Chinese archives (pp. 2 and 12). As her bibliography indicates, she has used the BM Archives sparingly and I cannot see what Chinese archives she has consulted. Much of the discussion follows the official mission historiography by Steiner and, above all, Schlatter. The author would thus do well to situate and tone down her claims. She should also take more information from the more recent research literature, which she is familiar with and which is in many ways superior to Schlatter.

9.     Some presentational issues. The author renders Chinese personal names in the Western fashion, with the family name last, but should use the Chinese convention (it is very odd to read Fuguang Xu for Xu Fuguang). In terms of place names, she uses both Hanyu Pinyin and missionary transliterations – where using the latter, she should always add the place name in Pinyin, perhaps with a reference how the place is called today (many countries and places changed their names after 1912). The author seems to have consulted some Chinese literature but has translated all bibliographical information into English. This is not the convention among China scholars. The author should list all bibliographical information in Chinese (Pinyin as standard, adding Chinese characters if editorial policy allows). Reference are missing on p. 3 (end of third paragraph; Hamberg) and p. 4 (end of second paragraph; letter from Stockholm), p. 5 (2nd paragraph from bottom, Hakka proverb).

10.  Finally, some typos, mistranslations and questionable word choices:

“babble” (p. 3) fantasy, or chimera, or exaggeration

“durch” (p.4 n. 5) dadurch (better delete the entire footnote with the quotation from the German)

“Dong Jing Dynasty (909)” (p. 5) Eastern Jin dynasty (ruled from 317 to 420 so the year 909 must be wrong; at least clarify which dynasty is meant here)

“Huangchao Rebellion (893–909)” Huang Chao Rebellion (874–883)

“Tartar” (p. 5) Mongols

“Ernest Johann Eitel” (p. 5) Ernst Johann Eitel (should use the German form here, the correct English form is Ernest John Eitel but was never used in connection with BM – note that the name appears elsewhere in the article as Ernest Johanna Eitel, Johanna being a female name)

“tormented” (p. 6) burdened

“family Lin” (p. 8) Li family; Zhenggao Lin Li Zhenggao

“Digler” (p. 9) Dilger

“Elisabeth” (p. 10) Oehler, or Oehler-Heimerdinger (Western convention is to use family not personal names)

Round 2

Reviewer 1 Report

I am very glad to see that the author improved his or her manuscript significantly. All the changes had been done in an appropriate manner, therefore I congratulate the author for his or her efforts in providing a very good article that, I am sure, will be very useful for many scholars in the field of the Christian missions in China. 

Reviewer 2 Report

Note that Hakka is usually not regarded as a dialect of Chinese but a Chinese language. The central government tries to impose a language policy that regards all Chinese languages as simply dialects, but this approach is politically motivated. 

Hakka and Mandarin are as far from each other as for instance German and English.